# Severe Scalp Psoriasis Microbiome Has Increased Biodiversity and Relative Abundance of Pseudomonas Compared to Mild Scalp Psoriasis

**DOI:** 10.3390/jcm11237133

**Published:** 2022-11-30

**Authors:** Jin-Young Choi, Hyunseong Kim, Ha-Yeh-Rin Koo, Jaeyoon You, Dong-Soo Yu, Young-Bok Lee, Minho Lee

**Affiliations:** 1Department of Dermatology, College of Medicine, The Catholic University of Korea, Seoul 06591, Republic of Korea; 2Department of Life Science, Dongguk University-Seoul, Ilsandong-gu, Goyang-si 10326, Republic of Korea

**Keywords:** bacteria, fungus, microbiome, psoriasis

## Abstract

Psoriasis is a chronic inflammatory skin disease associated with various factors. Recently, alterations in the gut and skin microbiomes have been shown to interact with host immunity, affect skin barrier function, as well as development and progression of psoriasis. We aimed to analyze the microbiota of the scalp of patients with psoriasis and determine the characteristics of the microbiome according to disease severity. We investigated the scalp microbiome of 39 patients with psoriasis scalp lesions and a total of 47 samples were analyzed. The patients were divided into mild, moderate, and severe groups according to the European recommendations for scalp psoriasis. For bacterial identification, we utilized the SILVA database targeting the V3 region of the 16 S rRNA gene. The mean Shannon index escalated along with disease severity, and the diversity of the scalp microbiome tended to increase with disease severity (R = 0.37, *p* < 0.01). The relative abundance of Pseudomonas was increased in severe scalp psoriasis (0.49 ± 0.22) compared to the mild group (0.07 ± 0.03, *p* = 0.029), and Diaphorobacter was enriched in the mild group (0.76 ± 0.16%) compared to the severe group (0.44 ± 0.22, *p* < 0.001). We identified that increased diversity of the scalp microbiome and the relative abundance of Pseudomonas are associated with the severity of scalp psoriasis.

## 1. Introduction

The human skin, the largest organ in the body, has a barrier function and a complex role in immunological interactions with the surrounding environment. One of the most important components of the environment is the microbial inhabitants of the skin, which are highly diverse and numerous and include bacteria, fungi, viruses, and mites [1]. Among these microorganisms, bacteria are known to help the host by inhibiting pathogen colonization and regulating T-cells. Conversely, they can also cause damage by producing virulence factors and invading the host immune system [2]. For example, exacerbation of atopic dermatitis is associated with staphylococcal colonization in lesioned skin [3].

Psoriasis is a chronic inflammatory skin disease affecting nearly 2–3% of the worldwide population, rendering it an important social, psychological, and economic burden [4,5]. Various genetic and environmental factors participate in disease initiation and exacerbation. Recently, the contribution of the skin and gut microbiota to the pathophysiology of psoriasis were highlighted. Streptococcal infection, as acute pharyngotonsillitis, can develop guttate psoriasis and exacerbate plaque psoriasis [6,7]. Similarly, several studies have found that the skin microbiome of psoriatic lesions is different from that of unaffected lesions or heathy controls [8]. For instance, *Streptococcus* on the skin of psoriatic lesions is more abundant than that on normal skin [9,10,11]. Unlike atopic dermatitis, microbiome diversity is higher in psoriatic plaque [9]. However, other studies have failed to reproduce these result [10]. Alterations in the microbiome can induce the interleukin 1 (IL-1) pathway, a key inflammatory factor in psoriasis [12]. Chang et al. reported that *Staphylococcus aureus* led to Th-17 polarization, one of the most crucial driving pathways in psoriasis [13].

In the last decade, the introduction of biological agents as a therapeutic strategy has initiated a new era in psoriasis treatment. Although more than half of the patients receiving biologics have experienced almost all clearing of psoriatic lesions, psoriatic plaques on the scalp tend to be resistant to therapy [14]. 

Scalp is involved up to 80% in psoriatic patients [15]. Due to its visibility, it can hugely affect the quality of life in a negative way [16]. It is reported that scalp involvement is one of the risk factors of severe psoriasis, increasing the risk by fivefold [17]. Although various treatments for psoriasis have been developed, there has not been an established treatment for scalp psoriasis yet; further, no treatment has been found to be sufficiently effective [18]. 

Considering that the scalp is one of the most frequently affected areas in patients with psoriasis, a new therapeutic method for recalcitrant scalp psoriasis is needed. There were a few studies on microbiome of scalp psoriasis, however, they are too small considering the unmet needs in the treatment of scalp psoriasis. Thus, we aimed to analyze the microbiota of the scalp of patients with psoriasis and determine the characteristics of the microbiome according to disease severity. By obtaining more perspectives on the microbiota in psoriasis, we could take one more step to understand the pathophysiology of psoriasis and facilitate tailored treatment options. 

## 2. Materials and Methods

### 2.1. Study Subjects and Sample Collection

We enrolled 39 patients diagnosed with scalp psoriasis and receiving treatment in the dermatology department of Uijeongbu St. Mary’s Hospital from September to November 2020. During the autumn of 2020 in Korea weather conditions were mild, with temperature ranging between 13.9 and 14.7 °C. Among them, we repeated the sample collection in 7 patients for a subgroup analysis. A total of 47 scalp microbiome samples were collected in the same room at the hospital by the same dermatologist. Written informed consent was obtained from all study subjects, which was approved by the Institutional Review Board of Uijeongbu St. Mary’s Hospital (UC20TESI0116). We stored samples in the Catholic Biobank and collected data on the patient’s sex, age, current treatment, and severity of the scalp lesions. After taking photographs of the whole body to calculate the PASI score, a sample from a psoriatic plaque on the patient’s scalp was obtained by swabbing the skin with a cotton swab placed into a sterile tube. 

### 2.2. Definition of PASI and Scalp PASI

Disease severity was assessed using the PASI [19]. The PASI, developed in 1978, evaluates the degree (0–4) of thickness, erythema, and scaling of the affected skin in four different body regions: head and neck, upper limbs, trunk, and lower limbs. The subtotal of each body site is then multiplied by the proportion of body surface area (0.1 for head and neck, 0.2 for upper extremities, 0.3 for a trunk, and 0.4 for lower extremities). By multiplying the area score ranging from 0 to 6, the total PASI can range from 0 to 72. The scalp PASI represents the regional PASI of the scalp and ranges from 0 to 7.2 [20]. The PASI score was analyzed as a continuous variable. We also divided the patients into three categories according to the European recommendations for the definition of mild, moderate, and severe scalp psoriasis [15]. The European recommendations measure the area affected by psoriatic plaques on the scalp (<50% or >50%), and the psoriatic features such as erythema, scaling, thickness, and pruritus. 

### 2.3. DNA Extraction and Sequencing

To extract bacterial DNA from the collected samples, a QIAamp DNA Mini spin column (Qiagen Co., Hilden, Germany,) was used. Tissue samples collected from the study participants were distributed into microcentrifuge tubes, which were centrifuged at 7500 rpm twice to obtain pellets. One hundred and eighty microliters ATL buffer and 20 μL proteinase K were mixed by vortexing, and each tube was incubated at 56 °C for one hour until the tissue was completely lysed. Subsequently, 200 μL AL buffer was added to the sample, followed by pulse-vortexing for 15 s and incubation at 70 °C for 10 min. Similarly, 200 μL ethanol (100%) was added to the sample, followed by pulse-vortexing for 15 s. 

The mixture, including the precipitate, was applied to a QIAamp Mini spin column (in a 2 mL collection tube) and centrifuged at 8000 rpm for 1 min. Subsequently, 500 μL AW1 buffer was added and the mixture was centrifuged at 8000 rpm for 1 min, followed by the addition 500 μL of AW2 buffer centrifugation at 14,000 rpm for 3 min. To eliminate the possibility of AW2 buffer carryover, each QIAamp Mini spin column was placed in a new 2 mL collection tube and centrifuged at 14,000 rpm for 1 min. The sample was incubated with 200 μL AE buffer at room temperature (20–22 °C) for 5 min, followed by centrifugation at 8000 rpm for 1 min. For further elution, the QIAamp Mini spin column was placed into a new tube, with the addition of 200 μL AE buffer, incubated and centrifuged once more.

Next, the V2–4 and V6–9 regions of 16 S rDNA were sequenced using the Ion S5™ XL System (Thermo Scientific, Waltham, MA, USA) and the primer set (Ion 16 s Metagenomics kit, Thermo Scientific) [21].

### 2.4. Bacterial Identification

All raw data were processed using the QIIME2 software (version 2021.4) [22]. The DADA2 [23] plugin for QIIME2 was used for primer removal, de-multiplexing, and feature-table generation. All reads were trimmed to 170 bp (--p-trunc-len 170). Reference sequences and classifiers were generated using the SILVA database (version 138) [24].

Alpha diversity values including Shannon’s index, Simpson’s index and the Chao1 index, and β-diversity were calculated based on the biological observation matrix (BIOM) table in QIIME2. Bray–Curtis dissimilarity and weighted UniFrac distances were used to analyze β-diversity. The β-diversity of the skin microbiome was visualized using a principal coordinate analysis (PCoA) [25,26,27].

Since the total abundance varied from sample to sample, we used relative abundance, which is a constant rather than absolute abundance. Relative abundance was calculated as the fraction of taxa observed in the feature table divided by the sum of all taxa within a sample.

### 2.5. In Silico Functional Analysis

Phylogenetic investigation of communities by reconstruction of unobserved states 2 (PICRUSt2, version 2.2.0-b) [28] was used to predict the functional potential of the microbiome in each sample based on the results of QIIME2. The PICRUSt2 results were generated for multiple classifications, such as Enzyme Commission (EC) numbers as well as Kyoto encyclopedia of genes and genomes (KEGG) orthologs and pathways. 

### 2.6. Statistical Analysis

All statistical analyses were performed using the R programming language (version 4.05). The Kruskal–Wallis test was used to compare more than two groups. The Wilcoxon rank-sum test was performed to compare two groups. Pearson correlation coefficients and Spearman’s rank correlation test were used to identify correlations between PASI and Shannon’s diversity index and were represented by R and ρ, respectively. 

## 3. Results

### 3.1. Demographics of the Scalp Psoriasis Patients

The 39 psoriatic patients who participated in this study had well-controlled psoriasis in areas other than the scalp. A total of 47 scalp psoriasis samples were collected in autumn, when weather conditions are mild in Korea (Table 1). The mean PASI including scalp PASI of 47 samples was 5.11 ± 4.52 suggesting that lesions had been controlled with the treatment which included biologics (*n* = 27), immunomodulators (*n* = 11), topical calcipotriol/betamethasone only (*n* = 6), and narrow band ultraviolet (UV) B (*n* = 2). The biologics used were secukinumab (*n* = 11), ustekinumab (*n* = 10), and guselkumab (*n* = 6). The mean scalp PASI scores were significantly different among the severity groups (*p* < 0.001). Among all samples the means were 1.71 ± 11.1, and 0.55 ± 0.24 for the mild, 1.59 ± 0.41 for the moderate, and 2.84 ± 0.82 for the severe group. Seven of the patients had psoriatic plaques only on the scalp. However, patients with higher scalp PASI scores had higher PASI on the whole body (mean PASI of 2.5 ± 1.42, 4.47 ± 4.84, and 7.99 ± 4.56, in the mild, moderate, and severe scalp psoriasis groups, respectively, *p* < 0.001).

### 3.2. Biodiversity According to the Severity of Scalp Psoriasis

Alpha diversity of scalp psoriasis was examined according to the severity of scalp psoriasis using the Shannon index (Figure 1a,b). The mean Shannon index was 0.87 ± 0.53, 1.18 ± 0.53, and 1.38 ± 0.34 in the mild, moderate, and severe group, respectively. A significant difference was observed between the groups (*p* = 0.02). The scalp PASI and Shannon entropy showed a statistically significant correlation (R = 0.37, *p* = 0.01, ρ = 0.41, *p* < 0.01) (Figure 1c). The PCoA using Bray–Curtis dissimilarity and weighted UniFrac showed that microbiome samples of mild scalp psoriasis were grouped closely and those of moderate and severe scalp psoriasis were similarly dispersed (Figure 2). 

### 3.3. Taxonomical Compositions of Each Severity Group

Figure 3 depicts the composition of the microbiota at the phylum and genus levels. Proteobacteria, Bacteroidetes, Firmicutes, and Actinobacteria were the most abundant phyla. The most abundant genus was *Diaphorobacter*, followed by *Pseudomonas*, *Staphylococcus*, *Cloacibacterium* and *Acinetobacter*. The relative abundance of *Diaphorobacter* in the mild group (0.76 ± 0.16) was significantly higher than that in the moderate (0.45 ± 0.25) and severe group (0.44 ± 0.22) (*p* < 0.001). *Pseudomonas* was more enriched in the severe group (relative abundance: 0.49 ± 0.22) than in the mild group (relative abundance: 0.07 ± 0.03) (*p* = 0.03) (Figure 4).

### 3.4. Functional Analysis

To predict bacterial gene function from the microbial metagenome, we used the PICRUSt2 algorithm. The heatmap yielded diverging patterns of KEGG orthology (KO) depending on the severity group (Figure 5a). Of the 67 KOs identified, 16 had different abundances between the severity groups (*p* < 0.01) (Appendix A). Among the 16 KOs that differed significantly depending on severity, five were associated with branched-chain amino acid transporters, two belonged to fatty acid metabolism, and one was linked to nitrogen metabolism. We also found an association between several bacterial taxa and the top 16 KOs (Figure 5b). K03704, a cold shock protein, diverged significantly more than other KOs among the severity groups (*p* < 0.0003) and showed an upward trend (Appendix A). This protein had a strong association with *Pseudomonas* which was most abundant in the severe group. Similarly, K07090, an uncharacterized protein, also had a strong correlation with *Pseudomonas* and increased in abundance with increasing severity (Appendix A). Other KOs related to metabolism had a relatively lower association with *Pseudomonas* than with *Diaphorobacter*. These exhibited a downward trend as the severity increased (Appendix A).

## 4. Discussion

The microbiome has been underestimated in the pathogenesis of scalp psoriasis. This study is the first analysis of microbiota focusing on the scalp in psoriasis patients, and it revealed differences in the microbiota depending on the severity of the disease.

Similar to a previous study by Drago et al. [11], Proteobacteria and Bacteroidetes were dominant at the phylum level in psoriatic skin. Conversely, Firmicutes was the most abundant phylum in other studies [9,10]. Assarsson et al. found that Firmicutes was significantly less abundant in psoriatic lesions than in healthy skin and decreased after UVB treatment [29]. 

In this study, the scalp PASI and diversity of the skin microbiome were correlated. This indicates that the microbial diversity increases when the disease is severe. Similarly, Gao et al. reported that the Shannon index in psoriatic lesions was significantly higher than that in healthy skin and controls [9]. In contrast, Alekseyenko et al. found that microbial diversity was much lower in patients with psoriasis than in those with other diseases [8]. This discrepancy may be partly due to differences in the sampling methods or sites. Further studies with larger populations and meta-analyses are required to elucidate this issue. 

*Pseudomonas* is well-known for its virulence and antibiotic resistance [30]. These pathogens can cause acute infections in the skin and eye, as well as chronic infections in the lung. In cystic fibrosis, the presence of *Pseudomonas aeruginosa* is associated with prognosis of the disease [31]. There is increasing evidence to support the link between *Pseudomonas* and psoriasis [13]. One study showed that *Pseudomonas* was more abundant in psoriatic plaques and decreased after treatment. Furthermore, there was no significant decrease in *Pseudomonas* abundance in patients who did not respond to treatment [29]. The authors speculated that topical vitamin D might lower the abundance of *Pseudomonas* [29]. We also found that *Pseudomonas* were enriched in the moderate and severe groups compared to the mild psoriasis group. In addition to psoriasis, *Pseudomonas* is associated with several systemic diseases. In multiple sclerosis, increased abundance of *Pseudomonas* in the gut microbiome has a pro-inflammatory effect [32]. An increased abundance of *Pseudomonas* has also been associated with inflammatory bowel disease and chronic rheumatic disease [33]. Although *Pseudomonas* is involved in the exacerbation of various inflammatory diseases, its exact mechanism and virulence factors are poorly understood.

In psoriatic skin, human β-defensin-2 is highly expressed in the stratum corneum, and its serum level correlates with disease severity [34]. *Pseudomonas* is a strong inducer of human β-defensin-2 in both skin and lung epithelia [35,36]. In addition to its antimicrobial activity, this antimicrobial peptide is likely an important part of Th17/22 cell recruitment in psoriasis pathogenesis [37]. Although the link between the skin microbiome and human β-defensin-2 is poorly understood, commensal bacteria may interact with this antimicrobial peptide, and compositional changes in the skin microbiome might influence the disease severity of psoriasis.

Coal tar is used to treat scalp psoriasis and is available in many forms from lotion to shampoo [38]. Although the exact mechanism is not thoroughly understood, it is known to have antiseptic and anti-inflammatory effects [39]. This product may affect microbial diversity and, therefore, exert a therapeutic effect. Additionally, antibiotics have been used to treat psoriasis assuming that the skin microbiome might worsen this disease [40]. Erythromycin produced antibacterial and anti-inflammatory effects, thus improving psoriasis. Macrolide antibiotics have a therapeutic effect on *Pseudomonas* and on streptococcal infections, which induce and aggravate psoriasis. Although antibiotic use for psoriasis treatment has not been studied in terms of anti-pseudomonal effects, antibiotics are possible candidates for future research on scalp psoriasis treatment.

The functional analysis revealed that five out of 16 KOs were associated with branched-chain amino acid transport. Branched-chain amino acids such as isoleucine, leucine, and valine are integral nutrient sources for bacteria. They not only help in protein synthesis, but also regulate the signals associated with amino acid starvation in bacteria [41]. In *Pseudomonas aeruginosa*, there are high-affinity branched-chain amino acid transporters encoded by *livJKHMF* that use ATP as an energy source [42]. However, the KO activities of branched-chain amino acid transporters were lower in the severe group than in the mild group, despite the higher relative abundance of *Pseudomonas* in the severe group. Similarly, KOs relevant to fatty acid metabolism and lipid biosynthesis had lower activity in the severe group than in the mild-to-moderate groups. Although it is unclear how the level of skin fatty acids changes due to bacterial metabolism, serum levels of free fatty acids correlate with psoriasis severity and amplify inflammation [43]. 

Interestingly, we repeated the sampling in 7 patients among 39 patients and found that there were large differences in composition in the same patients depending on disease severity. Although we could not find a statistical significance, when the PASI of a patient improved during treatment, the bacterial composition largely changed. When a patient’s disease status changed from severe to mild, either the abundance of *Pseudomonas* or Shannon index decreased. We did not find an association between age, sex, or treatment method and scalp microbiome composition.

This is the first analysis of the microbiome in scalp psoriasis according to disease severity. We analyzed 16 s RNA to identify the bacterial microbiota, which has been proven accurate in previous studies [44]. The homogeneous ethnicity of the participants is another strength of this study, although further studies of various ethnic groups are required. 

This study had several limitations. First, we did not include healthy controls. However, we categorized the groups according to the scalp PASI score and compared disease severity and biodiversity of the scalp microbiome. Second, there were no data on possible confounders such as antibiotic use. Third, the PASI evaluation can have subjective aspects. However, in this study, the PASI was evaluated by the same dermatologist. Fourth, we included the patients receiving various treatments. Although it is not known, the immunomodulation of each drug can affect the microbiome differently. Since the effects of the drug cannot be eliminated, it would be reasonable to include a group that does not get medication. Finally, the sample size of this study was small. This is because samples were collected for a short period in autumn since the skin microbiome is influenced by weather changes. Further studies with larger sample sizes are required to get a more definite conclusion.

In conclusion, the present study revealed differences in the microbiome between the severity groups of patients with scalp psoriasis. We found that skin microbiome of severe scalp psoriasis is significantly different from that of mild and moderate psoriasis, showing increased microbial diversity and relative abundance of *Pseudomonas.* Further studies are required to determine how the microbiome affects the disease and how to apply these findings towards therapeutic approaches.

## Figures and Tables

**Figure 1 jcm-11-07133-f001:**
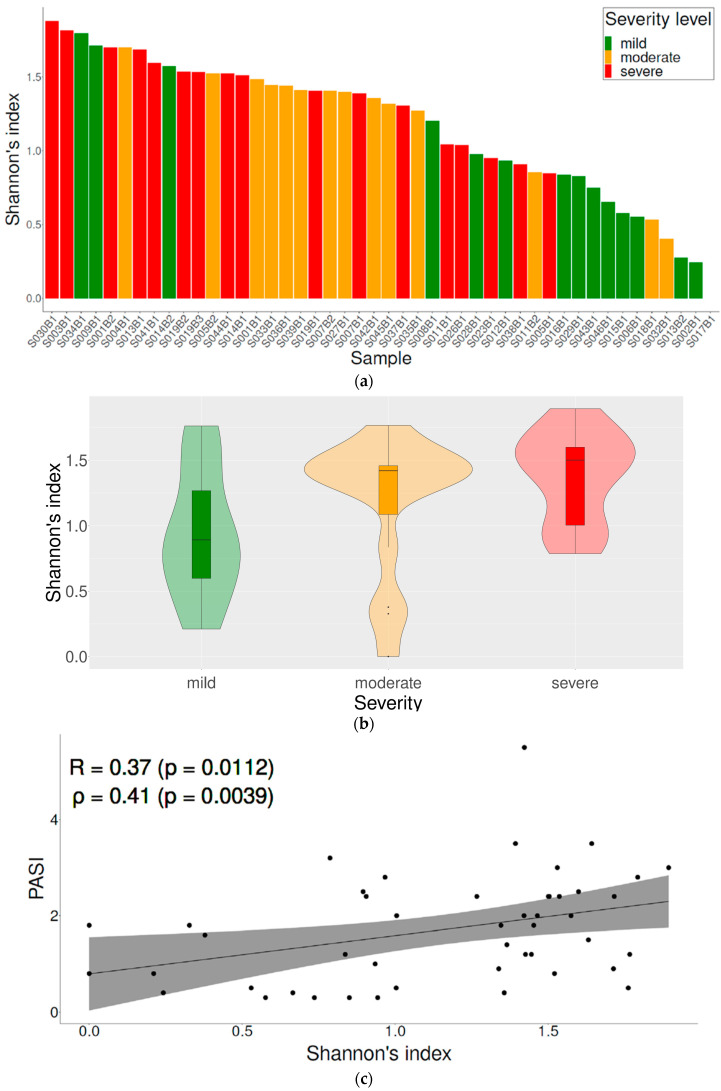
Relationship between severity of scalp psoriasis and biodiversity. (**a**) Shannon diversity index of each sample; (**b**) violin plot according to the severity; (**c**) correlation between PASI and Shannon diversity. PASI, psoriasis area and severity index; R, Pearson correlation coefficient; ρ, Spearman’s rank correlation test.

**Figure 2 jcm-11-07133-f002:**
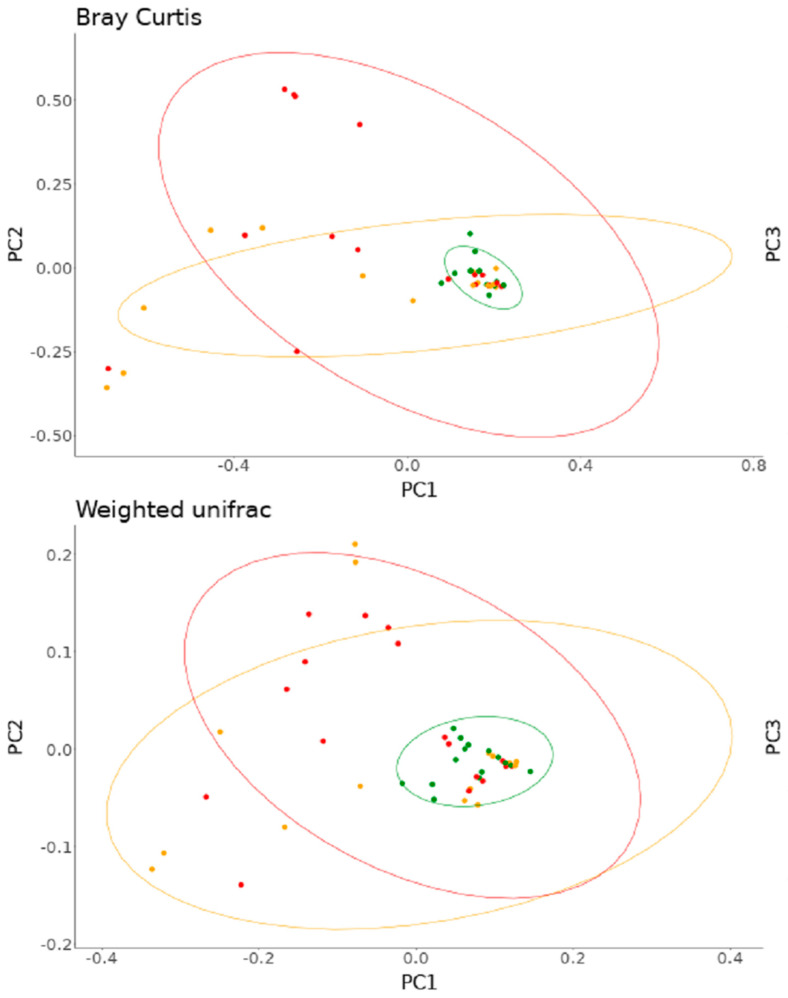
(**upper**) and Weighted UniFrac (**lower**). Green, orange, red dots denote mild, moderate, severe groups, respectively.

**Figure 3 jcm-11-07133-f003:**
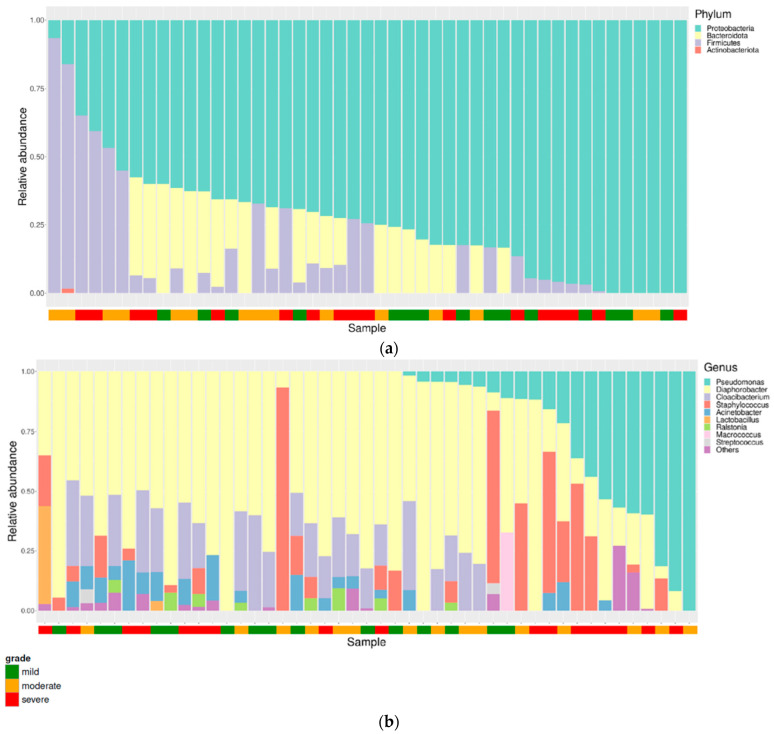
Composition of the microbiota at the (**a**) phylum and (**b**) genus levels. Green, orange, red dots denote mild, moderate, severe groups, respectively.

**Figure 4 jcm-11-07133-f004:**
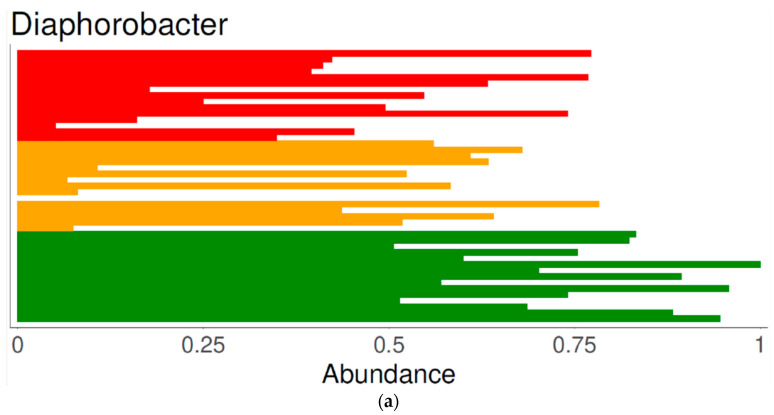
Relative abundance of genus Diaphorobacter (**a**) and Pseudomonas (**b**) in each sample and according to disease severity (**c**). Diaphorobacter was most abundant in the mild group of scalp psoriasis and Pseudomonas was most abundant in the severe group of scalp psoriasis. **: *p* < 0.01, ***: *p* < 0.001 and ns: *p* > 0.05.

**Figure 5 jcm-11-07133-f005:**
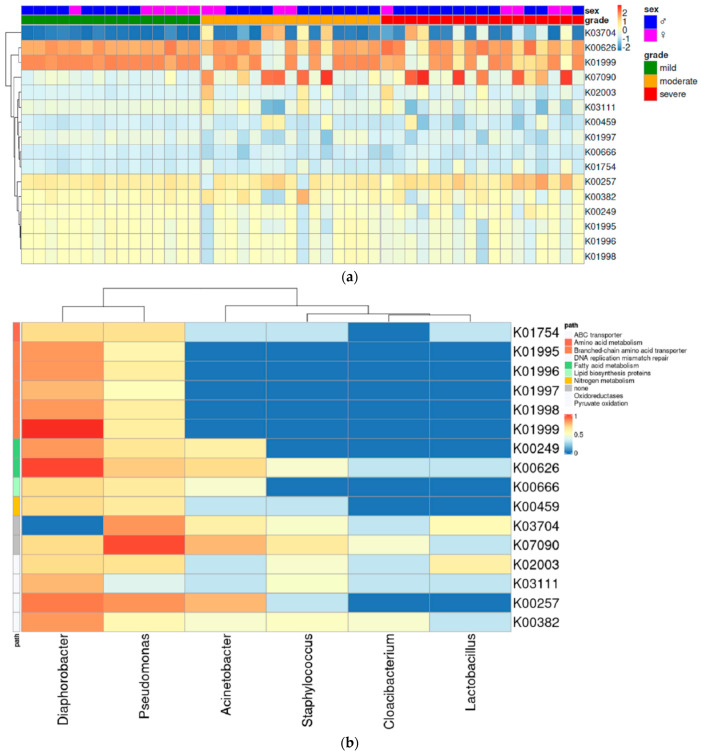
Bacterial gene functions were predicted using the PICRUSt2 algorithm. (**a**) Heatmap of KEGG orthology (KO) shows functional differences between the mild, moderate, and severe scalp psoriasis. (**b**) A total of 16 KOs differed significantly between groups (*p* < 0.01). They are related to several bacterial taxa. KEGG, Kyoto encyclopedia of genes and genomes; PICRUSt2, Phylogenetic investigation of communities by reconstruction of unobserved states 2.

**Table 1 jcm-11-07133-t001:** Characteristics and PASI of the patient in each severity groups.

Samples	Total (*n* = 47)	Mild (*n* = 15)	Moderate (*n* = 15)	Severe (*n* = 17)	*p*-Value
Age, mean (SD)	39.72 (12.86)	41.8 (14.0)	40.8 (13.12)	36.95 (11.82)	0.56
Gender (Male:Female)	33:14	9:6	11:4	12:5	0.71
Scalp PASI, mean (SD)	1.71 (1.10)	0.55 (0.24)	1.59 (0.41)	2.84 (0.82)	<0.001
PASI, mean (SD)	5.11 (4.52)	2.5 (1.42)	4.47 (4.84)	7.99 (4.56)	<0.001

PASI, psoriasis area and severity index; SD, standard deviation.

## Data Availability

The data that support the findings of this study are available from the corresponding authors.

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
