# Peer review of "Severe Scalp Psoriasis Microbiome Has Increased Biodiversity and Relative Abundance of Pseudomonas Compared to Mild Scalp Psoriasis"

_jcm, 2022, doi:10.3390/jcm11237133_

Round 1
Reviewer 1 Report
Authors studied scalp microbiome in psoriasis (PS) patients and found that PS severity is associated with increased diversity and abundance of pseudomonas. This is interesting, but there are several issues including originality, study design, and data interpretation.
1) PS involves multiple areas such as elbows, knees, scalp, low back, face and palms. Are there any specific reasons that authors focused on scalp microbiome? Are there any unique microbiome findings in scalp compared to other skin locations? Did authors compare microbiome profiles between scalp and other areas?
2) In lines 54-55, this is not true because microbiome in scalp of PS patients has been studied before. It has been known that pseudomonas spp are increased in PS skin (PMCID: PMC8069836, PMCID: PMC6125946, PMID: 31247199).
3) Microbiome samples were collected from 7 psoriatic patients and were analyzed based on severity of disease. This means just 2-3 subjects were used for each group, so this is not enough subjects to compare each group although authors collected multiple samples from a same subject. Therefore, the sample size is too small to get a conclusion as authors mentioned in the limitation of this study.
4) There is no good control group for this study since authors did not collect samples from normal healthy subjects. This is also a critical problem as authors mentioned in the limitation of this study.
Reviewer 2 Report
Dear Authors,
The study is interesting and gives new information about the difficulties in understanding this disease. There are not many studies on this subject and there is a great lack of knowledge about the impact of microbioma in psoriasis.
1. Since there is shortage of knowledge of microbioma impact on psoriasis; why did you not study patients without treatment?
2. The patients had well-controlled psoriasis in areas other than the scalp. Do the microbioma differ between the scalp and the rest of the body in psoriasis lesions from the same patient?
3. You choosed to enroll patients with different immunotherapies. Do you think that the treatment could have any impact on your results?
4. Can different immunoregulating treatments have impact on microbiome expression in situ?
5. What did you find most interesting with the study and why?
Round 2
Reviewer 1 Report
There are still concerns since the number of study subjects are too low (n=7) and healthy control samples are not included.